# Embroid: Unsupervised Prediction Smoothing Can Improve Few-Shot Classification

**Neel Guha\***
Stanford University

**Mayee F. Chen\***
Stanford University

**Kush Bhatia\***
Stanford University

**Azalia Mirhoseini**
Anthropic

**Frederic Sala**
University of Wisconsin-Madison

**Christopher Ré**
Stanford University

## Abstract

Recent work has shown that language models' (LMs) prompt-based learning capabilities make them well suited for automating data labeling in domains where manual annotation is expensive. The challenge is that while writing an initial prompt is cheap, improving a prompt is costly—practitioners often require significant labeled data in order to evaluate the impact of prompt modifications. Our work asks whether it is possible to improve prompt-based learning *without* additional labeled data. We approach this problem by attempting to modify the predictions of a prompt, rather than the prompt itself. Our intuition is that accurate predictions should also be consistent: samples which are similar under some feature representation should receive the same prompt prediction. We propose EMBROID, a method which computes multiple representations of a dataset under different embedding functions, and uses the consistency between the LM predictions for neighboring samples to identify mispredictions. EMBROID then uses these neighborhoods to create additional predictions for each sample, and combines these predictions with a simple latent variable graphical model in order to generate a final corrected prediction. In addition to providing a theoretical analysis of EMBROID, we conduct a rigorous empirical evaluation across six different LMs and up to 95 different tasks. We find that (1) EMBROID substantially improves performance over original prompts (e.g., by an average of 7.3 points on GPT-JT), (2) also realizes improvements for more sophisticated prompting strategies (e.g., chain-of-thought), and (3) can be specialized to domains like law through the embedding functions.

## 1 Introduction

Acquiring labeled data for domains like medicine and law is essential to training machine learning models or performing basic data analysis (e.g., "how many contracts contain a choice-of-forum clause" or "how many patient medical histories discuss an adverse reaction to a drug?") [15, 18, 19]. However, building large labeled datasets is difficult, and efforts like [25] show that manual labeling with domain experts is cost-prohibitive. Recent works have begun exploring if language models (LMs) could learn annotation tasks *in-context* [6] and replace manual labeling at scale [13, 15, 23, 31]. The promise of this approach is that LMs' in-context capabilities enable them to learn tasks from descriptions of the the task (i.e., *prompts*). However, the challenge is that producing high performance

---

*These authors contributed equally to this work.

37th Conference on Neural Information Processing Systems (NeurIPS 2023).

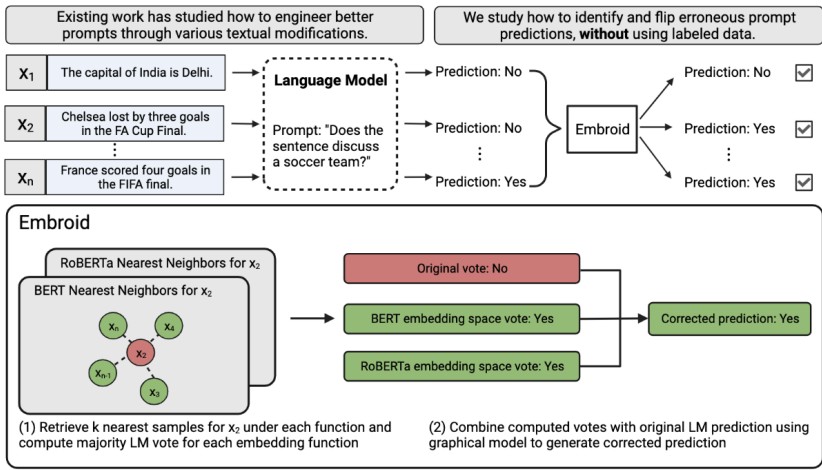

Figure 1: The EMBROID method for prompt-patching.

prompts is still expensive, as practitioners require labeled data in order to measure the impact of modifications to a prompt [51]. Existing work has thus focused on how domain experts can optimally construct prompts for a task (*prompt-engineering*), while minimizing reliance on labeled data [41, 58, 64]. Yet, because language models are sensitive to even small changes in prompt language, these techniques are imperfect and still produce erroneous predictions [2, 8, 38, 51, 70].

Our work approaches the challenge of improving prompt performance without labels from an orthogonal perspective: given the predictions of any prompted LM, can we identify and correct mis-predictions using *unlabeled* data? We describe this as the problem of *prompt-patching*. In the context of data annotation tasks, prompt-patching methods should meet three goals. First, they should be theoretically explainable, so that practitioners can understand when and how to apply them. Second, they should be fast, so that practitioners can efficiently integrate them into existing workflows. Finally, they should rarely be wrong, so that they don't worsen the performance of predictions.

Our work presents EMBROID: a method for automatically identifying and correcting LM predictions with unlabeled data and no expert supervision. Recent work has shown that for many tasks, samples close-by in embedding spaces (produced by models like BERT) have the same label [7]. EMBROID applies this intuition to the prompt-patching regime. Specifically, after LM predictions for all samples have been generated, EMBROID retrieves the $k$ most similar samples for each input, under $N$ different embedding functions. For each embedding function, EMBROID computes a scaled-modified majority vote over the LM's predictions for the $k$ retrieved samples. EMBROID then combines these $N$ votes with the original LM prediction for the test sample using a simple latent variable graphical model that is learned with a fast method-of-moments estimator [14]. The intuition behind EMBROID is that good prompts are *smooth* with respect to their predictions over a dataset—samples which are proximate under an embedding function should receive consistent predictions. Thus, modifying the predictions of a prompt to increase neighborhood agreement can improve the accuracy of those predictions. Lastly, because a single embedding space may imperfectly capture similarities between samples, retrieving neighbors from multiple embedding spaces improves robustness [28, 40].

Because EMBROID relies on weak-supervision—the subject of recent rigorous study [7]—it is possible to theoretically analyze and explain *why* and *when* EMBROID will improve performance. In particular, we find that performance is a function of the quality of the embeddings and the performance of the initial prompt. We also empirically study EMBROID, conducting experiments over six LMs, on up to 95 tasks, with several different prompt strategies. We find that EMBROID rarely worsens performance, and often improves F1 by a substantial margin. For instance, EMBROID improves GPT-3.5 by an average of 4.9 points F1 per task, and GPT-JT by an average of 7.3 points per task. The magnitude of EMBROID's gains are such that it enables a 1.3B parameter model to outperform an instruction-tuned 6.7B parameter model. EMBROID is also complementary to advanced prompt engineering strategies, and achieves performance improvements when applied to prompts designed

using chain-of-thought [64], AMA [2], and selective annotation [58]. Finally, EMBROID can be extended to specialized domains like law, through the use of already-available domain specific embeddings.

Succinctly, our contributions in this paper are: (1) EMBROID, a simple prompt-patching framework for improving LM predictions over text classification tasks; (2) a theoretical analysis of EMBROID which explains performance improvements in terms of embedding smoothness and base accuracy; and (3) an empirical evaluation of EMBROID covering up to 95 tasks and six different LMs.

## 2 Related work

**Improving LM performance** Improving the in-context generalization abilities of LMs has been intensely studied. The first family of approaches focuses on adapting LMs in order to make them more amenable to prompting. This includes task-specific finetuning [26, 27, 47], training on instruction data [9, 59], RLHF [50], and weight-surgery methods which attempt to "correct" incorrect information stored in model weights [10, 22, 43, 44]. A second family of approaches explores strategies for optimizing prompts to models, either through the specific textual features of the prompt [29, 45, 64], the use of task decompositions or LM recursion [2], implicit prompt representations [36, 37], or external databases [46]. Prompt-patching, in contrast, focuses on identifying mistakes in the predictions generated from a particular prompt. The most related approaches are aggregation methods, in which the outputs of multiple prompts are combined with an ensembling method [2, 41]. We find that EMBROID outperforms many such baselines, and can be applied to enhance their outputs.

**Weak supervision** EMBROID leverages statistical techniques developed in the weak supervision literature. The objective in weak supervision is to generate probabilistic labels for unlabeled data by combining the predictions of multiple noisy heuristics [14, 53, 54, 57, 62, 66]. EMBROID's novelty is that it uses embeddings to construct additional synthetic predictions, which are combined with the original predictions. In contrast, recent weak supervision approaches which incorporate embeddings use them to produce more fine-grained accuracy parameters [7], detect and discard training points [34], and as the basis for label propagation with final weak supervision predictions [52].

## 3 Problem setup and background

**Problem setup** Our problem setup comprises three elements: an unlabeled dataset, predictions from a LM for each sample in this dataset, and embedding representations of our dataset. Our goal is to improve the accuracy of LM predictions, by using the embedding representations to identify and correct predictions likely to be incorrect. Because recent work has explored how predictions from multiple prompts can be combined for a task [2], we present a generalized version of EMBROID in which we have access to multiple LM predictions. In our empirical evaluation however, we show that EMBROID performs well regardless of the number of predictions per sample available.

More formally, we focus on a binary classification task where $x \in \mathcal{X}$ denotes a sentence or paragraph and $y \in \mathcal{Y} = \{-1, 1\}$ is the binary label. We assume we are given an unlabeled dataset $\mathcal{D} = \{x_i\}_{i=1}^{n_u}$ of $n_u$ points. Each point $x$ is sampled i.i.d. from a distribution $\mathbb{P}_x$, and there exists a true underlying distribution $\mathbb{P}$ on the joint $(x, y)$. Following the true few-shot regime [51], we assume the only labels available are those used in the prompt. We denote a language model (e.g., GPT-3) as $\lambda_{\mathsf{LLM}}$, and a task-specific prompt as $\phi$, which prepends task instructions to input $x$ (e.g., "Does the clause contain an audit provision? Yes or No."). We consider the "prompt" to include both the task description and the in-context samples, consistent with [70]. The prediction this prompt induces for $\lambda_{\mathsf{LLM}}$ over $x$ is $\lambda_{\mathsf{LLM}}(\phi(x)) \in \mathcal{Y}$.[1] Varying $\phi$ by changing the task description, in-context demonstrations, or punctuation will alter the prediction generated for $x$. For a set of $m$ prompts $[\phi_1, \ldots, \phi_m]$, we denote their respective predictions on $x$ as a vector of *weak sources* $\boldsymbol{\lambda}(x) = [\lambda_{\mathsf{LLM}}(\phi_1(x)), \ldots, \lambda_{\mathsf{LLM}}(\phi_m(x))]$. For convenience, we denote $\lambda_{\mathsf{LLM}}(\phi_i(x))$ as $\lambda_i(x)$ or $\lambda_i$ when the $x$ is obvious, and similarly use $\boldsymbol{\lambda}$ instead of $\boldsymbol{\lambda}(x)$. We distinguish between two regimes: in the *single-prompt* regime with $m = 1$, we have access to a LM prediction for each point $x$, while in the *multi-prompt* regime with $m > 1$, we have access to multiple predictions.

---

[1]We assume a task-specific mapping function which allows a practitioner to associate a text generation from an LM to a particular class prediction in $\mathcal{Y}$.

**Algorithm 1** EMBROID: Correcting LLMs with embeddings

---

**Input:** Unlabeled data $\mathcal{D}$, LLM predictions $\boldsymbol{\lambda}(x)$ for each $x \in \mathcal{D}$, embedding models $\mathcal{E} = \{E_1, \ldots, E_N\}$, shrinkage parameter $\tau$, nearest neighbors parameter $k$
**for all** unlabelled $x \in \mathcal{D}$ **do**
    **for all** embedding models $E_j \in \mathcal{E}$ **do**
        Compute k-nearest neighbors $\mathrm{NN}_{j,k}(x)$
        Compute smoothed neighborhood prediction $\lambda_{\mathsf{sm},j}(x)$ using $\boldsymbol{\lambda}$, $\mathrm{NN}_{j,k}(x)$, and $\tau$ using eq. (2)
    **end for**
**end for**
Solve graphical model $\Pr(y, \boldsymbol{\lambda}(x), \boldsymbol{\lambda}_{\mathsf{sm}}(x))$ in eq. (3) with triplet method over $\mathcal{D}$ (Algorithm 2).
**for all** unlabeled $x \in \mathcal{D}$ **do**
    Sample $\hat{y}_x \sim \hat{\Pr}(y | \boldsymbol{\lambda}(x), \boldsymbol{\lambda}_{\mathsf{sm}}(x))$
**end for**
**Output:** Label set $\hat{\boldsymbol{Y}} = \{\hat{y}_x \mid x \in \mathcal{D}\}$

---

We assume access to $N$ embedding models $\mathcal{E} = [E_1, \ldots, E_N]$, each represented as a fixed mapping $E_i : \mathcal{X} \mapsto \mathcal{Z}_i$ from an input $x$ to an embedding vector $z$. These auxiliary embedding models provide representations of $x$ which encode different types of similarity information. Through model repositories like HuggingFace [65], it is possible to download a number of models which generate representations for text sentences (e.g., BERT or RoBERTa [12, 42]). These models have the property that semantically similar sentences are close-by in embedding space [7, 28, 40].

**Weak supervision background**    Weak supervision uses a graphical model to combine votes from multiple noisy sources into a single prediction, by estimating the accuracy of each source. It models $\Pr(y, \boldsymbol{\lambda}(x))$ as a latent variable graphical model and uses $\hat{y} = \mathrm{argmax}_y \hat{\Pr}(y | \boldsymbol{\lambda}(x))$ to produce label estimates, where $\hat{\Pr}$ represents the learned model. The graphical model is based on a graph $G = (V, E)$, where $V = y \cup \boldsymbol{\lambda}$ and $E$ consists of edges from $y$ to each $\lambda_j$. We assume no dependencies between sources, although simple extensions can incorporate them [61]. The formal graphical model is:

$$\Pr(y, \boldsymbol{\lambda}(x)) = \frac{1}{Z} \exp\big(\underbrace{\theta_y y}_{(I)} + \underbrace{\theta^\top \boldsymbol{\lambda}(x) y}_{(II)}\big) \tag{1}$$

where $Z$ is the partition function used for normalization, $(I)$ represents a label balance term with parameter $\theta_y$ controlling the prior of $\Pr(y = 1)$, and $(II)$ represents the source accuracy term where each $\theta_i$ is an *accuracy parameter* for the $i$th source. Note that from this model, sources are conditionally independent: $\lambda_i \perp\!\!\!\perp \lambda_j | y$ for any $i, j \in [m]$. Our use of this model has two steps. First, we must learn the accuracy parameters of $\Pr(y, \boldsymbol{\lambda}(x))$ without access to $y$. We use the triplet method introduced in [14], which is an efficient method-of-moments estimator for the parameters. Then, at inference we compute $\hat{\Pr}(y | \boldsymbol{\lambda}(x))$. Appendix B contains more details.

## 4   EMBROID

First, EMBROID uses the embedding models $\mathcal{E}$ to compute additional votes for each $x$. Let $\mathrm{NN}_{j,k}(x) \subset \mathcal{D}$ be the $k$-nearest neighbors of sample $x$ under the embedding function $E_j$. We define the smoothed neighborhood prediction vector $\lambda_{\mathsf{sm},j}(x) \in \{-1, 0, 1\}^m$ as follows, with $\lambda_{\mathsf{sm},j}[i](x)$ being the $i$th element:

$$\tilde{\lambda}_j[i](x) = \frac{1}{k} \sum_{\tilde{x} \in \mathrm{NN}_{j,k}(x)} \lambda_i(\tilde{x})$$

$$\lambda_{\mathsf{sm},j}[i](x) = \begin{cases} 1 & \tilde{\lambda}_j[i](x) > \tau_i^+ \\ -1 & \tilde{\lambda}_j[i](x) < \tau_i^- \\ 0 & \text{o.w.} \end{cases}, \tag{2}$$

where $\tau_i^+ \in [-1, 1]$ and $\tau_i^- \in [-1, 1]$ act as shrinkage parameters for $\lambda_i$ which control the level of agreement amongst the neighbors of $x$ necessary to generate a particular vote. The scalar $\lambda_{\mathsf{sm},j}[i](x)$ is the average vote of $\lambda_i$ amongst the neighbors of $x$ in $E_j$. When $\lambda_{\mathsf{sm},j}[i](x)$ is sufficiently positive, i.e., $\lambda_{\mathsf{sm},j}[i](x) > \tau_i^+$, EMBROID sets $\lambda_{\mathsf{sm},j}[i](x)$ to be a positive vote. When $\lambda_{\mathsf{sm},j}[i](x)$ is sufficiently negative, i.e., $\lambda_{\mathsf{sm},j}[i](x) < \tau_i^-$, EMBROID sets $\lambda_{\mathsf{sm},j}[i](x)$ to be a negative vote. Otherwise, $\lambda_{\mathsf{sm},j}[i](x)$ is set to be an abstain. The intuition is that $\lambda_{\mathsf{sm},j}[i](x)$ will be an accurate vote over $x$ whenever two conditions are met: (1) the LM is generally accurate, i.e., $\lambda_j$ is usually correct, and (2) $E_j$ is *smooth*, i.e., nearest-neighbors share the same task label.

Next, we augment our base model in equation (1) to incorporate these auxiliary neighborhood predictions $\boldsymbol{\lambda}_{\mathsf{sm}} = [\lambda_{\mathsf{sm},1}, \ldots, \lambda_{\mathsf{sm},N}] \in \{-1, 0, 1\}^{Nm}$ computed using the embeddings:

$$\Pr(y, \boldsymbol{\lambda}, \boldsymbol{\lambda}_{\mathsf{sm}}) = \frac{1}{Z} \exp\left(\theta_y y + \theta^\top \boldsymbol{\lambda} y + \sum_{j=1}^{N} \alpha_j^\top \lambda_{\mathsf{sm},j} y\right), \tag{3}$$

where the vector $\alpha_j \in \mathbb{R}^m$ represents the quality parameters for the $j^{th}$ embedding model when used with the $m$ different prompts. To solve this model and produce label estimates, we note that it has the same format as (1) if we concatenate $\boldsymbol{\lambda}$ and $\boldsymbol{\lambda}_{\mathsf{sm}}$ into one set of weak sources. Therefore, we can use the triplet method from [14] to learn parameters and output estimates $\hat{\Pr}(y|\boldsymbol{\lambda}(x), \boldsymbol{\lambda}_{\mathsf{sm}}(x))$ for each $x \in \mathcal{D}$ at inference time (see Appendix B for details).

Parameters $\theta$ and $\alpha_j$ in (3) allow us to trade-off two different sources of information—one presented by directly prompting an LM to obtain a label and the other by incorporating similarity information from the embedding models—and to further account for varying error modes among the embedding models. Our use of the neighborhood predictions in (3) yields a more expressive model than the standard weak supervision framework solely on LLM predictions in (1), which we can recover when $k = 0$, and can thus help make corrections to the LLM predictions. In practice, we find that setting $\tau_i^+ = \tau_i^- = \mathbb{E}[\lambda_i]$ (i.e., the average source vote) yields good performance (Appendix G).

## 5 Theoretical analysis

We analyze EMBROID, discussing the advantages of using $\boldsymbol{\lambda}_{\mathsf{sm}}$ in addition to $\boldsymbol{\lambda}$, and show that embedding smoothness and base prediction accuracy play a critical role in information gain. Appendix F provides synthetics demonstrating these tradeoffs and comparing to weak-supervision baselines.

First, we provide a result on the generalization error of our model $\hat{\Pr}(y|\boldsymbol{\lambda}, \boldsymbol{\lambda}_{\mathsf{sm}})$. Define the generalization error as the expected cross-entropy loss, $L(\boldsymbol{\lambda}, \boldsymbol{\lambda}_{\mathsf{sm}}, \mathcal{D}) = \mathbb{E}_{y, \boldsymbol{\lambda}(x), \boldsymbol{\lambda}_{\mathsf{sm}}(x), \mathcal{D}}[-\log \hat{\Pr}(y|\boldsymbol{\lambda}(x), \boldsymbol{\lambda}_{\mathsf{sm}}(x))]$. We use $[\lambda_1, \ldots, \lambda_{(N+1)m}]$ to represent $[\boldsymbol{\lambda}, \boldsymbol{\lambda}_{\mathsf{sm}}]$ and denote by $a_{\max} = \max_i \mathbb{E}[\lambda_i(x)y]$ the largest accuracy (scaled to $[-1, 1]$) of any source, and by $b_{\min} = \min_{i,j}\{\mathbb{E}[\lambda_i \lambda_j], \hat{\mathbb{E}}[\lambda_i \lambda_j]\}$ the minimum expected pairwise product between any two sources. Assume that all sources are better than random, e.g., $\Pr(\lambda_i = y) > 0.5$. These terms and assumptions are from using the triplet method.

**Proposition 5.1.** *Suppose that the data* $x, y, \boldsymbol{\lambda}, \boldsymbol{\lambda}_{\mathsf{sm}}$ *follows the model in* (3). *The generalization error of* $\hat{\Pr}(y|\boldsymbol{\lambda}, \boldsymbol{\lambda}_{\mathsf{sm}})$ *can be decomposed into*

$$L(\boldsymbol{\lambda}, \boldsymbol{\lambda}_{\mathsf{sm}}, \mathcal{D}) \leq \underbrace{H(y|\boldsymbol{\lambda}, \boldsymbol{\lambda}_{\mathsf{sm}})}_{\textit{Irreducible Error}} + \underbrace{\frac{C(N+1)m}{n_u}}_{\textit{Variance}} + o(1/n_u),$$

*where* $C = \frac{3(1-b_{\min}^2)}{8b_{\min}^2(1-a_{\max}^2)}\left(\frac{1}{b_{\min}^4} + \frac{2}{b_{\min}^2}\right).$

In the bound above, the variance term comes from estimation error when learning the parameters via the triplet method. The irreducible error depends on quality of $\boldsymbol{\lambda}$ and $\boldsymbol{\lambda}_{\mathsf{sm}}$. If knowledge of the LLM prediction and neighborhood prediction significantly reduces uncertainty in $y$, the conditional entropy term $H(y|\boldsymbol{\lambda}(x), \boldsymbol{\lambda}_{\mathsf{sm}}(x))$ is low.

**Information gain from using both $\boldsymbol{\lambda}, \boldsymbol{\lambda}_{\mathsf{sm}}$** We compare upper bounds on generalization error when both $\boldsymbol{\lambda}, \boldsymbol{\lambda}_{\mathsf{sm}}$ are modeled versus when only $\boldsymbol{\lambda}$ is modeled, as in (1) corresponding to classical weak

supervision. Based on the bound in Proposition 5.1, modeling both $\boldsymbol{\lambda}$ and $\boldsymbol{\lambda}_{\mathsf{sm}}$ increases the variance term by a constant multiplicative factor.

Here, we examine how the irreducible error is affected, that is, the difference $H(y|\boldsymbol{\lambda}) - H(y|\boldsymbol{\lambda}, \boldsymbol{\lambda}_{\mathsf{sm}})$. Since this quantity is always nonnegative, we focus on bounding the *pointwise* difference in conditional entropy—which we call the information gain—for a given $x_0$ on which the LLM is incorrect. For simplicity, suppose we have one embedding $E$. An embedding $E$ is *M-smooth* with respect to the label if

$$\Pr(\tilde{y} = c | y = c, \|E(x) - E(\tilde{x})\| \leq \varepsilon) \geq M_E(\varepsilon), \tag{4}$$

where $c \in \mathcal{Y}$, $\varepsilon > 0$ and $M_E(\cdot) \in [0, 1]$ is decreasing in its input. Note that this definition requires knowledge of ground-truth labels, and is thus impossible to use as a metric for selecting which embeddings to use.

Define $\beta_i = \Pr(\lambda_i = y)$ as the accuracy of $\lambda_i$ and $p_{\boldsymbol{\lambda}} = \Pr(y = 1|\boldsymbol{\lambda}(x_0))$ as the prediction on $x_0$ given only access to $\boldsymbol{\lambda}$. Let $\varepsilon_k = \max_{\tilde{x} \in \mathrm{NN}_k(x)} \|E(x) - E(\tilde{x})\|$ be the maximum distance between $x_0$ and its $k$ neighbors. Without loss of generality, assume the label on $x_0$ is $y = 1$.

**Theorem 5.2.** *Assume that $E$ is $M$-smooth. The pointwise information gain on $x_0$ is*

$$H(y|\boldsymbol{\lambda}(x_0)) - H(y|\boldsymbol{\lambda}(x_0), \boldsymbol{\lambda}_{\mathsf{sm}}(x_0)) \geq$$

$$2(1 - p_{\boldsymbol{\lambda}}) \left[ \prod_{i=1}^{m} \left[ 1 - \exp[-2k(\beta_{NN_k,i} - 0.5)^2] \right] - 0.5 \right]$$

*where $\beta_{NN_k,i} = \Pr_{\tilde{x} \sim NN_k}(\lambda_i(\tilde{x}) = y) \geq \beta_i M_E(\varepsilon_k)$ is the neighborhood accuracy.*

A few observations on the bound are in order.

- **Improvement over WS:** If the neighborhood accuracy is bounded sufficiently far from $\frac{1}{2}$ and $k$ is large, using EMBROID has better irreducible error than just using $\boldsymbol{\lambda}$. For example, setting $m = 1$, $k = 10$, $\beta_{\mathrm{NN}_k,i} = 0.7$, and $p_{\boldsymbol{\lambda}} = 0.25$ gives us an improvement of 0.076 nats.

- **Smoothness:** If $E$ is highly smooth, then $M_E(\varepsilon_k)$ will be large and irreducible error will be small.

- **Base prediction accuracy:** If the original sources $\boldsymbol{\lambda}$ have high accuracy ($\beta_i$), irreducible error will be small.

Additionally, we observe that if $p_{\boldsymbol{\lambda}}$ is a high-quality prediction close to the true label 1, the information gain is small. Choice of the $k$ parameter presents a performance trade-off: increasing $k$ will increase $\varepsilon_k$ and incorporate farther-away, less reliable predictions, but it will also reduce the noise of the majority vote. We also comment on the information gain when using both $\boldsymbol{\lambda}$ and $\boldsymbol{\lambda}_{\mathsf{sm}}$ over just $\boldsymbol{\lambda}_{\mathsf{sm}}$ in Appendix C.

## 6 Results

Our empirical evaluation focuses on three questions: (1) How robust is EMBROID's performance across LMs? (2) How does EMBROID, as a prompt-patching method, compare to high performance prompt-engineering methods? (3) How sensitive is EMBROID to the embeddings and dataset size?

**Tasks** We study tasks where sentence embeddings can capture information relevant to the task, leading us to focus on sentence classification datasets. We consider a collection of 95 class-balanced sentence classification tasks, derived from binarizing existing multi-class legal, scientific, and general domain classification benchmarks like CUAD, AGNews, DBpedia-14, FewRel, and several others [21, 25, 30, 67, 69].[2] Example tasks include, "Classify if the following texts discuss a recording label" or "Classify if the following contractual clauses contain an audit rights provision."

---

[2]We hope to explore multi-class extensions and more complex reasoning tasks in future work.

|  | LM | Win rate (%) | Avg. Improvement (F1) |
|---|---|---|---|
| API-Access Models | J1-Jumbo (176B) | 72.2 | 10.6 |
| | GPT-3.5 (> 170B) | 80.6 | 4.9 |
| Open Source | Bloom (7.1B) | 91.2 | 10.1 |
| | OPT (6.7B) | 91.2 | 11.6 |
| Instruction Tuned | GPT-JT (6B) | 89.1 | 7.3 |

Table 1: We evaluate the extent to which EMBROID improves the original prompt on different models in terms of win rate and relative improvement (defined in-line). All models are run with three trials. For each model, we report the percentage of tasks (across all trials) for which EMBROID improves, and the average improvement (in F1 points). Additional details provided in Appendix.

**Choice of embedding models**    Following prior work illustrating the benefits of domain specific representation [20, 71], EMBROID uses different embeddings for each task domain. For law tasks, we rely on two BERT-variants trained on different legal corpora [24, 71]. For science tasks, we rely on three BERT-variants trained on science, biology, and medical texts [3, 17, 35]. For general domain tasks, we rely on BERT, Roberta, and SentenceBert embeddings [12, 42, 55].

**Prompts**    Prompts are constructed using fixed instructions, and by manually selecting three random samples (from each class) as in-context demonstrations (Appendix E). We follow the true few-shot regime [51], in that we assume the only labeled data available to the data scientist are the labels used for in-context demonstrations. Prior work has found this regime to most realistically represent real-world workflows.

**Models**    We evaluate on two API-access models: GPT-3.5 (`text-davinci-003`) and J1-Jumbo [39]. Because API models raise significant privacy and compliance concerns for data scientists working with sensitive data [16], we also evaluate on open-source models. We select models in the 6-7B parameter range, as these are the largest models which fit on commonly available 40GB A100 machines. Specifically, we evaluate Bloom [56] and OPT [68]. Given the increasing popularity of instruction-tuning, we also evaluate on GPT-JT [60], an 6.7B parameter instruction tuned version of GPT-J. Because of cost-constraints, we evaluate API-access models on a representative selection of 12 tasks, while evaluating all other models on the full suite of 95 tasks. Appendix D provides details.

## 6.1   By how much does prompt-patching improve performance?

**Performance across LM families**    We examine if EMBROID achieves improvements for different *types* of LMs. For each LM, we select three different combinations of in-context demonstrations (i.e., three prompts), generate predictions for each prompt, and apply EMBROID to independently each prompt's predictions. This produces $3 \times 95 = 285$ trials for open-source models, and $3 \times 12 = 36$ trials for API-models. We report *win-rate*, i.e., the proportion of trials for which EMBROID outperforms the original predictions, and *improvement*, i.e., the average difference in F1 points (across all trials) between EMBROID and the original predictions.

As Table 1 illustrates, EMBROID improves performance for a substantial proportion of prompts, by a substantial margin, across all models. On GPT-3.5 for instance, EMBROID achieves a win-rate of 80.6%, with an average of improvement of 4.9 points. EMBROID also improves for open source models, with a win-rate of 91.2% on OPT-6.7 and an average improvement of 11.6 points. Finally, EMBROID achieves similar gains on an instruction tuned model, with a win-rate of 89.1% and an average improvement of 7.3 points.

**Performance when prompts are good**    We additionally investigate how EMBROID's performance improvements change as a function of the performance of the base prompt. Hypothetically, one could imagine that better performing prompts are *smoother* with respect to embeddings, thus diminishing (or negating) EMBROID. In Figure 2 (upper left), we plot the improvement of EMBROID against the performance of the base prompt for GPT-JT. Even when the base prompt performs well (i.e., F1 > 0.8), EMBROID improves on 89% of tasks by an average of 4.1 points.

| LM | MV | Liger | FlyingSquid | AMA | EMBROID-1 | EMBROID-3 |
|---|---|---|---|---|---|---|
| J1-Jumbo | 47.4 | 48.7 | 50.5 | 60.7 | 60.4 | **64.5** |
| GPT-3.5 | 81.4 | 82.5 | 82.1 | 84.7 | 83.9 | **86.0** |
| Bloom-7.1B | 54.6 | 55.8 | 54.3 | 63.0 | 64.7 | **69.1** |
| OPT-6.7 | 46.1 | 46.8 | 46.3 | 56.3 | 59.8 | **64.2** |
| GPT-JT | 69.3 | 69.4 | 70.1 | 74.6 | 75.1 | **79.0** |

Table 2: We evaluate how EMBROID compares to common ensemble approaches for improving prompt prediction performance. All ensemble baselines are run with three sets of predictions. EMBROID-1 is run with one set of predictions, and EMBROID-3 is run with three set of predictions. For each method, we report the average macro-F1 over all tasks. We observe that EMBROID-1 is competitive with ensemble methods which use many more predictions, while EMBROID-3 outperforms all other methods by a substantial margin.

**Measuring performance in parameter count**   A trend in recent literature has been to measure the magnitude of improvements to prompt performance in terms of parameter count [2], by showing how a particular method makes a smaller method equivalent in performance to a larger model. We find that EMBROID enables the non-instructed tuned 1.3B GPT-Neo model to outperform an instruction tuned 6.7B model; across all trials, GPT-JT scores an average F1 of 67.8, while EMBROID +GPT-Neo-1.3B scores an average of 68.5.

## 6.2   Comparing prompt-patching to prompt-engineering

Our work distinguishes between prompt-construction methods—which control how a prompt is generated—and prompt-patching methods—which attempt to identify and correct errors in the predictions produced by a prompt. We use EMBROID to further study the difference between these frameworks in two ways. First, we compare EMBROID's performance improvement over a base prompt to that of several specialized prompting strategies. Second, we examine the extent to which EMBROID—when applied to the predictions produced by these prompting strategies—can generate further performance improvements. We study three prompting strategies:

1. Ensemble strategies, in which the predictions of multiple prompts are combined using an unsupervised ensembling model. Specifically, we compare to two ensembling methods previously studied for LLMs (AMA [2] and majority vote [41]), one ensembling method which incorporates embedding information (Liger [7]), and one well regarded weak supervision baseline (FlyingSquid [14]). Each baseline is run over the predictions generated by three different prompts.

2. Chain-of-thought prompting [64], in which for each in-context demonstration, we provide a step-by-step explanation for the demonstration's label.

3. Selective annotation (SA) [58], in which we use embeddings to select a subset of $k$ data samples to label, and then, for each input sample, retrieve the most similar samples (under some embedding function) from this pool to use as in-context demonstrations.

**Ensemble methods**   We evaluate two versions of EMBROID. In the first version, we run EMBROID with the predictions of only one prompt (EMBROID-1). In the second version, we run EMBROID with the predictions of three different prompts (EMBROID-3). Note that this requires performing inference over the few-shot LM three times for each sample, which can be expensive. This baseline is comparable to applying EMBROID to the outputs of an ensemble method. In Table 2, we observe that EMBROID-1 is competitive with the ensemble baselines (while using substantially fewer predictions), while EMBROID-3 consistently outperforms these baselines (across different LMs).

**Chain-of-thought**   We compare EMBROID to chain-of-thought (CoT) prompting for a subset of a representative subset of 10 tasks on GPT-3.5. For each task, we manually construct a base prompt consisting of six demonstrations, and a CoT prompt where an explanation is provided for each demonstration. We first find that EMBROID's performance improvement over the base prompt exceeds that of chain-of-thought prompting (Table 3). Using EMBROID to modify the base prompt is better

| Base prompt | +CoT | +EMBROID | + CoT + EMBROID |
|---|---|---|---|
| 76.3 | 80.1 | 81.9 | **85.4** |

Table 3: We evaluate EMBROID compared to, and applied to, CoT prompting on GPT-3.5 for a subset of 10 tasks. We report the average across the studied tasks.

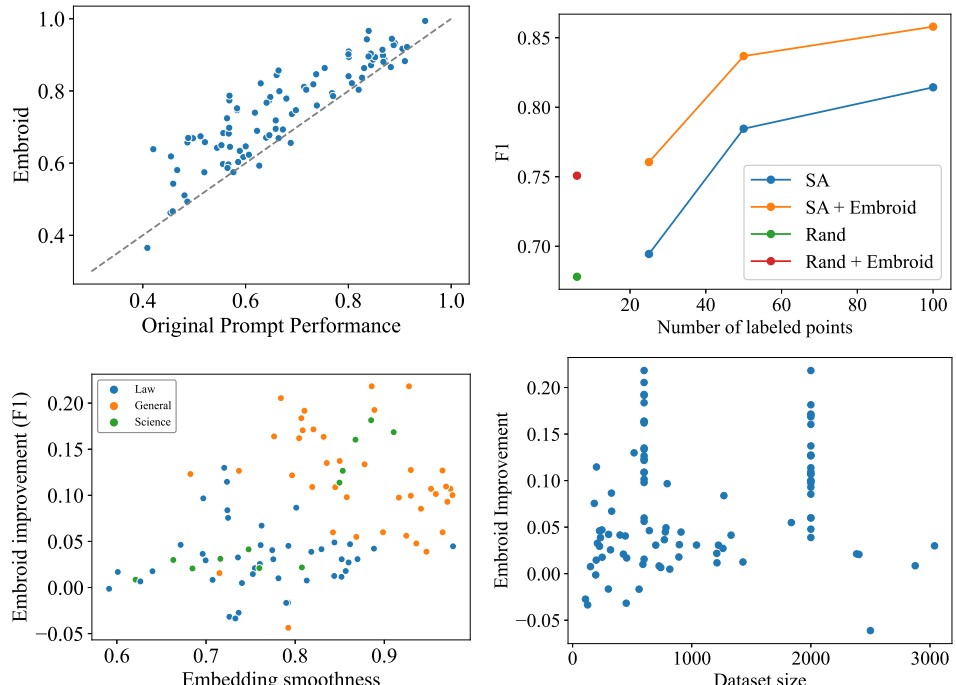

Figure 2: **Upper left**: The EMBROID F1 plotted against the F1 score for the original prompt for GPT-JT. Even for high performing prompts, EMBROID is capable of improving performance. The dashed line $y = x$ is plotted for visual aid. **Upper right**: A comparison of EMBROID to selective annotation (SA) over all tasks for GPT-JT. **Bottom left**: For each task (using GPT-JT), we plot the performance improvement of EMBROID against the average smoothness of the embeddings used. We observe a positive correlation ($r = 0.39$). **Bottom right**: Across all tasks, we measure the performance improvement of EMBROID against the size of the task.

on average than CoT prompting, outperforming CoT on six out of the ten tasks. Second, applying EMBROID to predictions generated by a CoT prompt yields further improvements, outperforming vanilla CoT predictions on eight of ten tasks.

**Selective annotation (SA)**  We compare EMBROID to selective annotation with a label budget of $[6, 25, 50, 100]$ (Figure 2, upper-right). For each task, we run selective annotation using a domain specific embedding. EMBROID (applied to a prompt with randomly chosen samples) outperforms selective annotation with a label budget of 25 samples. When a label budget of 100 samples is available, EMBROID improves the performance of a prompt constructed using selective annotation on 88% of tasks, by an average of 4.3 points.

### 6.3 Ablations

Finally, we perform several ablations of EMBROID to study how performance changes as a function of (1) the domain specificity of the embedding used, (2) the quality of the embedding spaces used, and (3) the size of the dataset. Additional ablations are presented in Appendix G.

**Domain specific embeddings improve performance** We compare how performance on the legal and science tasks changes when we shift from domain specialized embeddings to general domain embeddings. On law tasks for GPT-JT, we find that using two legal embedding spaces outperforms using BERT and RoBERTa for 77% of tasks, by up to 6 points F1 on certain tasks [24, 71]. For science tasks for GPT-JT, we find that using two science embedding spaces [3, 35] outperforms using BERT and RoBERTa for 92% of tasks, by up to 4.3 points F1 on certain tasks.

**Embedding quality** Building on Section 5, we compare EMBROID's performance improvement over the base prompt to the average smoothness of the embedding spaces with respect to each task (Figure 2). We observe a positive correlation: smoother embedding spaces are associated with larger performance gains (with a Pearson coefficient of $r = 0.39$). Applying this insight, we explore how performance changes when *extremely* high quality embeddings are added. For a subset of 19 tasks we generate OpenAI `text-embedding-ada-002` embeddings, and find that adding them to EMBROID improves performance by up to 13 points F1 (at an average of 2 points across all studied tasks).

**Dataset size** Finally, we study how EMBROID's performance improvement changes as the dataset size changes. Because EMBROID relies on nearest-neighbors in different embedding spaces, we might expect performance to be poor when the dataset being annotated is small. In Figure 2 (bottom right), we see that EMBROID achieves performance improvements even for "small" datasets with only several hundred samples. ⁻

## 7 Conclusion

We study the problem of improving prompt-based learning, by developing a method (EMBROID) for detecting and correcting erroneous predictions without labeled data. We validate EMBROID across a range of datasets and LMs, finding consistent improvement in many regimes. We take a moment to address the societal impact of our work: while we do not foresee any *direct* harmful impacts arising from our work, we caution that any use of language models in meaningful applications should be accompanied by conversations regarding risks, benefits, stakeholder interests, and ethical safeguards.

## 8 Acknowledgements

We are grateful to Arjun Desai, Avanika Narayan, Benjamin Spector, Dilara Soylu, Gautam Machiraju, Karan Goel, Lucia Zheng, Rose E. Wang, Sabri Eyuboglu, Sarah Hooper, Simran Arora, Maya Varma, and other members of the Hazy Research group for helpful feedback and conversations on this project.

We gratefully acknowledge the support of NIH under No. U54EB020405 (Mobilize), NSF under Nos. CCF1763315 (Beyond Sparsity), CCF1563078 (Volume to Velocity), 1937301 (RTML) and CCF2106707; US DEVCOM ARL under No. W911NF-21-2-0251 (Interactive Human-AI Teaming); ONR under No. N000141712266 (Unifying Weak Supervision); ONR N00014-20-1-2480: Understanding and Applying Non-Euclidean Geometry in Machine Learning; N000142012275 (NEP-TUNE); NXP, Xilinx, LETICEA, Intel, IBM, Microsoft, NEC, Toshiba, TSMC, ARM, Hitachi, BASF, Accenture, Ericsson, Qualcomm, Analog Devices, Google Cloud, Salesforce, Total, the HAI-GCP Cloud Credits for Research program, the Stanford Data Science Initiative (SDSI), the Wisconsin Alumni Research Foundation (WARF), the Center for Research on Foundation Models (CRFM), and members of the Stanford DAWN project: Facebook, Google, and VMWare. The U.S. Government is authorized to reproduce and distribute reprints for Governmental purposes notwithstanding any copyright notation thereon.

Any opinions, findings, and conclusions or recommendations expressed in this material are those of the authors and do not necessarily reflect the views, policies, or endorsements, either expressed or implied, of NIH, ONR, or the U.S. Government.

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

# A Notation

The glossary is given in Table 4 below.

| Symbol | Used for |
|---|---|
| $x$ | Input sentence or paragraph $x \in \mathcal{X}$. |
| $y$ | Binary task label $y \in \mathcal{Y} = \{-1, +1\}$. |
| $\mathcal{D}$ | Unlabeled dataset $\mathcal{D} = \{x_i\}_{i=1}^{n_u}$ of $n_u$ points. |
| $\mathbb{P}, \mathbb{P}_x$ | The joint distribution of $(x, y)$ and the marginal on $x$, respectively. |
| $n_l$ | Number of labeled in-context examples used in querying the LLM (5-10 examples). |
| $\lambda_{\mathsf{LLM}}(\cdot), \phi(\cdot)$ | Users interact with a language model $\lambda_{\mathsf{LLM}}$ via a prompt $\phi$ on $x$. |
| $m$ | Number of prompts that we have access to. |
| $\boldsymbol{\lambda}$ | $\boldsymbol{\lambda}(x) = [\lambda_1(x), \dots, \lambda_m(x)]$ where $\lambda_i(x)$ is shorthand for $\lambda_{\mathsf{LLM}}(\phi_i(x))$. |
| $\mathcal{E}$ | The set of $N$ embedding models, $\mathcal{E} = \{E_1, \dots, E_N\}$ where each embedding is represented as a fixed mapping $E_i : \mathcal{X} \mapsto \mathcal{Z}_i$. |
| $Z$ | Partition function for normalization of (1). |
| $\theta_y, \theta$ | $\theta_y$ is a label balance parameter and each $\theta_i$ is a scalar accuracy parameter for the $i$th source in (1). |
| $\mathrm{NN}_{j,k}(x)$ | The $k$-nearest neighbors of $x$ in embedding space $E_j$. |
| $\boldsymbol{\lambda}_{\mathsf{sm}}$ | $\boldsymbol{\lambda}_{\mathsf{sm}}(x) = [\lambda_{\mathsf{sm},1}, \dots, \lambda_{\mathsf{sm},N}] \in \{-1, 0, 1\}^{Nm}$, where $\lambda_{\mathsf{sm},j} = [\lambda_{\mathsf{sm},j}[1], \dots \lambda_{\mathsf{sm},j}[m]]$ and $\lambda_{\mathsf{sm},j}[i](x)$ is the smoothed neighborhood prediction of $\lambda_i(x)$ in $E_j$ (eq. (2)). |
| $\tau_i^+, \tau_i^-$ | Shrinkage parameters for determining when $\lambda_{\mathsf{sm},j}[i](x)$ is set to 0, −1, or 1. |
| $\alpha_j$ | Vector of $m$ accuracy parameters for $E_j$ when used with $m$ prompts in (3). |
| $L(\boldsymbol{\lambda}, \lambda_{\mathsf{sm},}, \mathcal{D})$ | Generalization error of EMBROID ( expected cross-entropy loss). |
| $a_{\max}$ | The largest scaled accuracy of any source, $a_{\max} = \max_i \mathbb{E}\left[\lambda_i(x)y\right]$. |
| $b_{\min}$ | The smallest expected pairwise product between any two sources, $b_{\min} = \min_{i,j}\{\mathbb{E}\left[\lambda_i \lambda_j\right], \hat{\mathbb{E}}\left[\lambda_i \lambda_j\right]\}$. |
| $M_E(\cdot)$ | An embedding is $M$-smooth if $\Pr(\tilde{y} = c\|y = c, \|E(x) - E(\tilde{x})\| \leq \varepsilon) \geq M_E(\varepsilon)$ for all $c \in \mathcal{Y}$ and any $\varepsilon > 0$, where $M_E(\cdot) \in [0, 1]$ is decreasing in its input. |
| $\beta_i$ | The accuracy of $\lambda_i$, $\beta_i = \Pr(\lambda_i = y)$. |
| $p_{\boldsymbol{\lambda}}$ | The prediction on $x_0$ given only access to $\boldsymbol{\lambda}$, $p_{\boldsymbol{\lambda}} = \Pr(y = 1\|\boldsymbol{\lambda}(x_0))$. |
| $\varepsilon_k$ | The maximum distance between $x_0$ and its $k$ neighbors, $\varepsilon_k = \max_{\tilde{x} \in \mathrm{NN}_k(x)} \|E(x) - E(\tilde{x})\|$. |

Table 4: Glossary of variables and symbols used in this paper.

# B  Weak supervision background

In this section, we provide details on the inference and learning procedures for solving the graphical model defined in equation (1). The content from this section is derived from [14] and [7].

**Pseudolabel inference.** To perform inference, we compute $\hat{\Pr}(y|\boldsymbol{\lambda}(x))$ for some $x \in \mathcal{X}$. This is done via Bayes' rule and the conditional independence of weak sources:

$$\Pr(y = 1|\boldsymbol{\lambda}(x)) = \frac{\prod_{i=1}^{m} \Pr(\lambda_i(x)|y = 1)\Pr(y = 1)}{\Pr(\boldsymbol{\lambda}(x))}. \tag{5}$$

We assume that the class balance is known; for our datasets, the class balance is $\Pr(y = 1) = 0.5$. More generally, it can be estimated [53]. The latent parameter of interest in this decomposition is $\Pr(\lambda_i = 1|y = 1)$, which corresponds to the accuracy of $\lambda_i$.

---

**Algorithm 2** Triplet method [14]

---

**Input:** Dataset $\mathcal{D}$, weak sources $\boldsymbol{\lambda}(x)$.
**for** $i \in [m]$ **do**
    **for** $j, k \in [m] \backslash i$ **do**
        Estimate $\hat{\mathbb{E}}[\lambda_i \lambda_j]$ over $\mathcal{D}$, and similarly estimate $\hat{\mathbb{E}}[\lambda_i \lambda_k]$ and $\hat{\mathbb{E}}[\lambda_j \lambda_k]$.
        Compute $\hat{a}_i^{j,k} = \sqrt{\left|\frac{\hat{\mathbb{E}}[\lambda_i \lambda_j]\hat{\mathbb{E}}[\lambda_i \lambda_k]}{\hat{\mathbb{E}}[\lambda_j \lambda_k]}\right|}$.
    **end for**
    Calculate average $\hat{a}_i = \text{Mean}(\hat{a}_i^{j,k} \quad \forall j, k \in [m] \backslash i)$.
    Compute estimated accuracy $\hat{\Pr}(\lambda_i = y) = \frac{1+\hat{a}_i}{2}$.
**end for**
**Output:** Accuracies $\hat{\Pr}(\lambda_i = y)$ for all $i \in [m]$.

---

**Source parameter estimation: Triplet method.** Previous approaches have considered how to estimate $\Pr(\lambda_i = 1|y = 1)$ via the *triplet method* [14], which exploits conditional independence properties. First, by the properties of the graphical model in (1), it holds that the accuracy of $\lambda_i$ is symmetric: $\Pr(\lambda_i = 1|y = 1) = \Pr(\lambda_i = -1|y = -1) = \Pr(\lambda_i = y)$ (Lemma 4 of [7]). Therefore, $\Pr(\lambda_i = 1|y = 1)$ can be written in terms of $\mathbb{E}[\lambda_i y]$ with $\mathbb{E}[\lambda_i y] = 2\Pr(\lambda_i = 1|y = 1) - 1$.

Define $a_i = \mathbb{E}[\lambda_i y]$. The graphical model in (1) tells us that $\lambda_i y \perp\!\!\!\perp \lambda_j y$ if $\lambda_i \perp\!\!\!\perp \lambda_j|y$, which holds for all $i, j \in [m]$ (Proposition 1 of [14]). As a result, $\mathbb{E}[\lambda_i y] \times \mathbb{E}[\lambda_j y] = \mathbb{E}[\lambda_i \lambda_j y^2] = \mathbb{E}[\lambda_i \lambda_j]$, which is a quantity that can be computed from observed LLM predictions. That is, we have that $a_i a_j = \mathbb{E}[\lambda_i \lambda_j]$. If we introduce a third $\lambda_k$, we can generate a system of equations over $a_i, a_j, a_k$ in terms of their pairwise rates of agreements:

$$a_i a_j = \mathbb{E}[\lambda_i \lambda_j] \tag{6}$$
$$a_i a_k = \mathbb{E}[\lambda_i \lambda_k] \tag{7}$$
$$a_j a_k = \mathbb{E}[\lambda_j \lambda_k]. \tag{8}$$

Solving, we get that

$$|a_i| := \sqrt{\left|\frac{\mathbb{E}[\lambda_i \lambda_j]\mathbb{E}[\lambda_i \lambda_k]}{\mathbb{E}[\lambda_j \lambda_k]}\right|}, \tag{9}$$

and likewise for $a_j, a_k$. If we assume that each weak source is better than random over the dataset, then $a_i = |a_i| > 0$, so we can uniquely recover the accuracy of each source by selecting two other sources and computing the above expression by using empirical expectations over $\mathcal{D}$. We then set $\hat{\Pr}(\lambda_i = 1|y = 1) = \frac{1+\hat{a}_i}{2}$ and plug this into the expression for $\Pr(y = 1|\boldsymbol{\lambda}(x))$ in (5).

This approach is formally described in Algorithm 2.

## C   Proofs

### C.1   Proof of proposition 5.1

We note that $[\boldsymbol{\lambda}, \boldsymbol{\lambda}_{\mathsf{sm}}]$ can be viewed as a set of sources in the weak supervision set up used in [7, 14]. Therefore, we can apply Theorem 1 from [7] to our problem setting, noting that we do not perform their clustering step and that our predictions do not abstain and output 0 in addition to $\{-1, 1\}$. We have a total of $(N+1)m$ sources, so

$$L(\boldsymbol{\lambda}, \boldsymbol{\lambda}_{\mathsf{sm}}, \mathcal{D}) \leq H(y|\boldsymbol{\lambda}, \boldsymbol{\lambda}_{\mathsf{sm}}) + \frac{3(1 - b_{\min}^2)}{8b_{\min}^2(1 - a_{\max}^2)}\left(\frac{1}{b_{\min}^4} + \frac{2}{b_{\min}^2}\right)\frac{(N+1)m}{n_u} + o(1/n_u). \quad (10)$$

### C.2   Proof of theorem 5.2

We can write the change in point-wise irreducible error as follows:

$$H(y|\boldsymbol{\lambda}(x_0)) - H(y|\boldsymbol{\lambda}(x_0), \boldsymbol{\lambda}_{\mathsf{sm}}(x_0)) = \mathbb{E}\left[-\log \Pr(y|\boldsymbol{\lambda}(x_0)) + \log \Pr(y|\boldsymbol{\lambda}(x_0), \boldsymbol{\lambda}_{\mathsf{sm}}(x_0))\right] \quad (11)$$

$$= \mathbb{E}\left[\log \frac{\Pr(y|\boldsymbol{\lambda}(x_0), \boldsymbol{\lambda}_{\mathsf{sm}}(x_0))}{\Pr(y|\boldsymbol{\lambda}(x_0))}\right] \quad (12)$$

$$= \mathbb{E}\left[\log\left(\frac{\Pr(\boldsymbol{\lambda}(x_0), \boldsymbol{\lambda}_{\mathsf{sm}}(x_0)|y)\Pr(y)}{\Pr(\boldsymbol{\lambda}(x_0), \boldsymbol{\lambda}_{\mathsf{sm}}(x_0))} \cdot \frac{\Pr(\boldsymbol{\lambda}(x_0))}{\Pr(\boldsymbol{\lambda}(x_0)|y)\Pr(y)}\right)\right] \quad (13)$$

$$= \mathbb{E}\left[\log \frac{\Pr(\boldsymbol{\lambda}_{\mathsf{sm}}(x_0)|\boldsymbol{\lambda}(x_0), y)}{\Pr(\boldsymbol{\lambda}_{\mathsf{sm}}(x_0)|\boldsymbol{\lambda}(x_0))}\right]. \quad (14)$$

Next, we use the fact that $\boldsymbol{\lambda}(x_0) \perp\!\!\!\perp \boldsymbol{\lambda}_{\mathsf{sm}}(x_0)|y$ to simplify the expression into

$$\mathbb{E}\left[\log \frac{\Pr(\boldsymbol{\lambda}_{\mathsf{sm}}(x_0)|y)}{\Pr(\boldsymbol{\lambda}_{\mathsf{sm}}(x_0)|y=1)\Pr(y=1|\boldsymbol{\lambda}(x_0)) + \Pr(\boldsymbol{\lambda}_{\mathsf{sm}}(x_0)|y=-1)\Pr(y=-1|\boldsymbol{\lambda}(x_0))}\right]. \quad (15)$$

The exact $\boldsymbol{\lambda}_{\mathsf{sm}}(x_0)$ is unknown but is drawn from the distribution $\Pr(\boldsymbol{\lambda}_{\mathsf{sm}}|y=1)$ since $x_0$'s label is 1. Then, this expression becomes an expectation over $\boldsymbol{\lambda}_{\mathsf{sm}}$:

$$\mathbb{E}_{\boldsymbol{\lambda}_{\mathsf{sm}}|y=1}\left[\log \frac{\Pr(\boldsymbol{\lambda}_{\mathsf{sm}}|y=1)}{\Pr(\boldsymbol{\lambda}_{\mathsf{sm}}|y=1)p_{\boldsymbol{\lambda}} + \Pr(\boldsymbol{\lambda}_{\mathsf{sm}}|y=-1)(1-p_{\boldsymbol{\lambda}})}\right]. \quad (16)$$

Given that our $\boldsymbol{\lambda}_{\mathsf{sm}}$ is high-quality, we suppose that $\boldsymbol{\lambda}_{\mathsf{sm}}(x_0)$ all equal 1 with high probability, and then we can lower bound our expression by

$$\Pr(\boldsymbol{\lambda}_{\mathsf{sm}} = 1|y = 1)\log \frac{\Pr(\boldsymbol{\lambda}_{\mathsf{sm}} = 1|y = 1)}{\Pr(\boldsymbol{\lambda}_{\mathsf{sm}} = 1|y = 1)p_{\boldsymbol{\lambda}} + \Pr(\boldsymbol{\lambda}_{\mathsf{sm}} = 1|y = -1)(1-p_{\boldsymbol{\lambda}})}. \quad (17)$$

The key quantity of interest is $\Pr(\boldsymbol{\lambda}_{\mathsf{sm}} = 1|y = 1) = \prod_{i=1}^m \Pr(\boldsymbol{\lambda}_{\mathsf{sm}, [i]} = 1|y = 1)$. We focus on bounding $\Pr(\boldsymbol{\lambda}_{\mathsf{sm}, [i]} = 1|y = 1)$ next. Suppose that the $k$ neighbors of $x_0$ are $x_1, \ldots, x_k$. Define $p_j = \Pr(\lambda_i(x_j) = 1|y = 1)$ for all $j \in [k]$. Note that $\lambda_i(x_j) \perp\!\!\!\perp \lambda_i(x_{j'})|y$ for any $j, j' \in [k]$ (while $\boldsymbol{\lambda}_{\mathsf{sm}, [i]}$ as a whole is dependent on $y$, individual neighbors are still conditionally independent). Then, the event that $\boldsymbol{\lambda}_{\mathsf{sm}, [i]} = 1|y = 1$ is as least as likely as the event that $\text{Binomial}(k, \min_{i \in [k]} p_i) \geq \frac{k}{2}$. Let $p_{\min} = \min_{i \in [k]} p_i$, and assume that $p_{\min} \geq \frac{1}{2}$. Then,

$$\Pr(\boldsymbol{\lambda}_{\mathsf{sm}, [i]} = 1|y = 1) \geq \Pr\left(\text{Binomial}(k, p_{\min}) \geq \frac{k}{2}\right) = \Pr\left(\frac{1}{k}\sum_{j=1}^k X_j \geq \frac{1}{2}\right) \quad (18)$$

$$= \Pr\left(\frac{1}{k}\sum_{j=1}^k X_j \geq p_{\min} - \left(p_{\min} - \frac{1}{2}\right)\right), \quad (19)$$

where $X_j \sim \text{Bernoulli}(p_{\min})$. Next, let $\delta = p_{\min} - \frac{1}{2}$. We can apply Hoeffding's inequality to get

$$\Pr\left(\text{Binomial}(k, p_{\min}) \geq \frac{k}{2}\right) = \Pr\left(\frac{1}{k}\sum_{j=1}^{k} X_j \geq p_{\min} - \delta\right) = 1 - \Pr\left(\frac{1}{k}\sum_{j=1}^{k} X_j \leq p_{\min} - \delta\right) \geq 1 - \exp(-2\delta^2 k) \tag{20}$$

$$= 1 - \exp(-2k(p_{\min} - 0.5)^2). \tag{21}$$

All that's left is to lower bound $p_{\min}$. Without loss of generality, suppose that $p_{\min}$ corresponds to an arbitrary $p_j = \Pr(\lambda_i(x_j) = 1|y = 1)$. We can decompose this probability into

$$\Pr(\lambda_i(x_j) = 1|y = 1) = \Pr(\lambda_i(x_j) = 1, y(x_j) = 1|y = 1) + \Pr(\lambda_i(x_j) = 1, y(x_j) = -1|y = 1) \tag{22}$$

$$= \Pr(\lambda_i(x_j) = 1|y(x_j) = 1)\Pr(y(x_j) = 1|y = 1) + \Pr(\lambda_i(x_j) = 1|y(x_j) = -1)\Pr(y(x_j) = -1|y = 1). \tag{23}$$

Since $\Pr(\lambda_i(x_j) = 1|y(x_j) = 1)$ is over all $x_j \sim \mathbb{P}_x$, this quantity is just equal to the accuracy of $\lambda_i$, $a_i$. Next, recall that $\|E(x_j) - E(x)\| \leq \varepsilon_k$, where $\varepsilon_k = \max_{x_i \in \text{NN}(x)} \|E(x) - E(x_i)\|$ is the maximum distance from the $k$ neighbors to $x$. Then, we can write $\Pr(y(x_j) = 1|y = 1)$ as $\Pr(y(x_j) = 1|y = 1, \|E(x_j) - E(x)\| \leq \varepsilon_k) \geq M_E(\varepsilon_k)$, since we have assumed that $E$ is $M$-smooth. We can now bound $p_{\min}$:

$$p_{\min} \geq a_i M_E(\varepsilon_k) + (1 - a_i)(1 - M_E(\varepsilon_k)). \tag{24}$$

Therefore, we have that

$$\Pr(\boldsymbol{\lambda}_{\text{sm}} = 1|y = 1) \geq \prod_{i=1}^{m} \left[1 - \exp[-2k(a_i M_E(\varepsilon_k) - 0.5)^2]\right]. \tag{25}$$

Before we plug in $\Pr(\boldsymbol{\lambda}_{\text{sm}} = 1|y = 1)$ into (17), we simplify the expression. Note that $\Pr(\boldsymbol{\lambda}_{\text{sm}} = 1|y = 1)$ can be written as $\prod_{i=1}^{m} p_i$ for some $p_i$, and $\Pr(\boldsymbol{\lambda}_{\text{sm}} = 1|y = -1)$ can be written as $\prod_{i=1}^{m}(1 - p_i)$. A simple proof by induction shows that $\prod_{i=1}^{m}(1 - p_i) \leq 1 - \prod_{i=1}^{m} p_i$. Therefore, we can write that (17) is lower bounded by

$$\Pr(\boldsymbol{\lambda}_{\text{sm}} = 1|y = 1)\log\frac{\Pr(\boldsymbol{\lambda}_{\text{sm}} = 1|y = 1)}{\Pr(\boldsymbol{\lambda}_{\text{sm}} = 1|y = 1)p_{\boldsymbol{\lambda}} + (1 - \Pr(\boldsymbol{\lambda}_{\text{sm}} = 1|y = 1))(1 - p_{\boldsymbol{\lambda}})} \tag{26}$$

Let's abbreviate $\Pr(\boldsymbol{\lambda}_{\text{sm}} = 1|y = 1)$ as $x$ and define the function

$$f(x) = x\log\frac{x}{xp_{\boldsymbol{\lambda}} + (1 - x)(1 - p_{\boldsymbol{\lambda}})}. \tag{27}$$

We note that for $x \geq 0.5$, $f(x)$ is convex and can thus be lower bounded by $f(x) \geq f'(0.5)(x - 0.5)$. We compute $f'(x) = \frac{1 - p_{\boldsymbol{\lambda}}}{xp_{\boldsymbol{\lambda}} + (1-x)(1-p_{\boldsymbol{\lambda}})}$, so $f'(0.5) = 2(1 - p_{\boldsymbol{\lambda}})$. Therefore, $f(x) \geq 2(1 - p_{\boldsymbol{\lambda}})(x - 0.5)$. Our final bound on the pointwise difference in irreducible error on $x_0$ is

$$H(y|\boldsymbol{\lambda}(x_0)) - H(y|\boldsymbol{\lambda}(x_0), \boldsymbol{\lambda}_{\text{sm}}(x_0)) \geq 2(1 - p_{\boldsymbol{\lambda}})\left[\prod_{i=1}^{m}\left[1 - \exp[-2k(a_i M_E(\varepsilon_k) - 0.5)^2]\right] - 0.5\right]. \tag{28}$$

**Information gain from using $\boldsymbol{\lambda}, \boldsymbol{\lambda}_{\text{sm}}$ over $\boldsymbol{\lambda}_{\text{sm}}$** We briefly comment on the opposite direction—how much does using both LLM predictions and neighborhood predictions help over just using neighborhood predictions?

The quantity we aim to lower bound is $H(y|\boldsymbol{\lambda}_{\text{sm}}(x_0)) - H(y|\boldsymbol{\lambda}(x_0), \boldsymbol{\lambda}_{\text{sm}}(x_0))$ for a point of interest $x_0$. We can write this quantity as

$$H(y|\boldsymbol{\lambda}_{\text{sm}}(x_0)) - H(y|\boldsymbol{\lambda}(x_0), \boldsymbol{\lambda}_{\text{sm}}(x_0)) = \mathbb{E}\left[\log\frac{\Pr(\boldsymbol{\lambda}(x_0)|y)}{\Pr(\boldsymbol{\lambda}(x_0)|\boldsymbol{\lambda}_{\text{sm}}(x_0))}\right] \tag{29}$$

Without loss of generality, suppose that the true label on $x_0$ is $y = 1$, and that for each $\lambda_i$, the neighborhood around $x_0$ consists of a balanced mix of $\lambda_i = 1$ and $\lambda_i = -1$. Then, with high probability we have that $\Pr(y = 1|\boldsymbol{\lambda}_{\mathsf{sm}}(x_0)) = p_{\boldsymbol{\lambda}_{\mathsf{sm}}} \approx 0.5$. From our proof of Theorem 5.2, we can thus write

$$H(y|\boldsymbol{\lambda}_{\mathsf{sm}}(x_0)) - H(y|\boldsymbol{\lambda}(x_0), \boldsymbol{\lambda}_{\mathsf{sm}}(x_0)) = \mathbb{E}_{\lambda(x_0)}\left[\log \frac{\Pr(\boldsymbol{\lambda}(x_0)|y = 1)}{\Pr(\lambda(x_0)|y = 1)p_{\boldsymbol{\lambda}_{\mathsf{sm}}} + \Pr(\lambda(x_0)|y = -1)(1 - p_{\boldsymbol{\lambda}_{\mathsf{sm}}})}\right] \tag{30}$$

$$\geq \Pr(\boldsymbol{\lambda}(x_0) = 1|y = 1))\log \frac{\Pr(\boldsymbol{\lambda}(x_0) = 1|y = 1)}{\Pr(\boldsymbol{\lambda}(x_0) = 1|y = 1)p_{\boldsymbol{\lambda}_{\mathsf{sm}}} + \Pr(\boldsymbol{\lambda}(x_0) = 1|y = -1)(1 - p_{\boldsymbol{\lambda}_{\mathsf{sm}}})} \tag{31}$$

$$\geq 2(1 - p_{\boldsymbol{\lambda}_{\mathsf{sm}}})(\Pr(\boldsymbol{\lambda}(x_0) = 1|y = 1) - 0.5) \tag{32}$$

If $\boldsymbol{\lambda}$ on $x_0$ has high accuracy and $p_{\boldsymbol{\lambda}_{\mathsf{sm}}}$ is low, then we can have significant point-wise information gain from modeling both $\boldsymbol{\lambda}$ and $\boldsymbol{\lambda}_{\mathsf{sm}}$ rather than just $\boldsymbol{\lambda}_{\mathsf{sm}}$.

# D   Datasets

**Motivation**. We study the performance of our method across a diverse collection of *task definitions*. In our setting, a task definition denotes a specific classification that a data scientist wishes to perform. For instance, a data scientist working on quantifying the breadth of legal issues that individuals face may wish to identify which posts in an online forum refer implicate legal issues related to housing.

This evaluation strategy is motivated by the observation that task definitions vary in their smoothness across embedding spaces, as different embeddings may do a better job of capturing features relevant for the task. For instance, out-of-the-box Sentence-BERT embeddings are better than traditional BERT at capturing the topicality of a sentence [55]. By focusing on a broad range of task definitions, we can better forecast how our method might perform for new tasks that practitioners may need to create classifiers for. We also avoid issues with leakage that may arise as the practice of finetuning LLMs on tasks increases [9].

In total, we study 95 distinct task definitions, encompassing 100,418 total samples. Each task varies between 108 and 3308 samples.

**Legal tasks**. The emergence of LLMs is exciting for law and finance, where expert-annotations are especially difficult to acquire [4, 18, 19]. Drawing on recent benchmarks and released datasets framing the potential use cases for LLMs in law, we study the following datasets:

- CUAD [25]: The original CUAD dataset consists of 500 contracts spanning an array of sectors, with clauses manually into one of 41 legal categories. Following [18], we adapt the original dataset for clause-by-clause classification. We turn each clause type into a binary classification task, where the objective is to distinguish clauses of that type from clauses of other types (i.e. "negatives"). Negative clauses are sampled randomly so as to make the task class balanced. We ignore clauses for which there are insufficient annotations in the original dataset.

- Learned Hands [33]: The Learned Hands dataset consists of legal questions that individuals publicly posted to an online forum (r/legaladvice). The questions have been coded by experts into legal categories according to the Legal Issues Taxonomy [32]. We consider several such issue classes, and create a binary classification task for each issue. Negative clauses are sampled randomly so as to make the task class balanced. Because these questions can be long, we truncate them at 50 tokens.

**Science tasks**. LLMs have generated similar excitement for medical and science informatics applications [1, 4]. We study established classification/extraction benchmarks.

- Chemprot [30]: ChemProt consists of sentences from PubMed abstracts describing chemical-protein relationships. We study seven relations, and create a binary task for each one. Each task is class balanced, with negatives sampled from the other relations.

- RCT [11]: The RCT dataset consists of PubMed abstracts for papers discussing randomized control trials, where sentences in the abstract are annotated according to their semantic role (e.g., background, methods, results, etc). There are five roles, and we create a binary task for each one. Each task is class balanced, with negatives sampled from the other relations.

**General domain tasks**. Finally, we study the following "general domain" tasks, derived from popular sentence classification and information extraction benchmarks.

- FewRel [21]: This is a relationship classification/extraction dataset, where each sample corresponds to a sentence mentioning the relationship between two entities. We select 20 relations, and for each relation construct a binary classification task with 700 positive instances of the relation, and 700 randomly sampled sentences (corresponding to other relations).

- Spam Detection [67]: We study the YouTube spam detection task from the WRENCH benchmark. This task requires classifying YouTube comments as spam/not spam.

- Toxic content detection [5]: This task requires classifying whether posted comments are toxic or not. We use a sampled subset of the CivilComments dataset.

- AG News [69]: The original dataset organizes news snippets into four categories: World, Sports, Business, and Science/Technology. We create a separate task for each category. Negatives are sampled from the remaining classes.
- DBPedia [69]: DBPedia is a 14-way ontology classification dataset. We convert this into 14 distinct tasks, corresponding to each of the ontology types.

Table 5: Legal tasks

| Task | Description/Intent | Size |
|---|---|---|
| Affiliate License-Licensee (CUAD) | Does the clause describe a license grant to a licensee (incl. sublicensor) and the affiliates of such licensee/sublicensor? | 208 |
| Anti-Assignment (CUAD) | Does the clause require consent or notice of a party if the contract is assigned to a third party? | 1212 |
| Audit Rights (CUAD) | Does the clause discuss potential audits? | 1224 |
| Cap On Liability (CUAD) | Does the clause specify a cap on liability upon the breach of a party's obligation? | 1262 |
| Change Of Control (CUAD) | Does the clause give one party the right to terminate if such party undergoes a change of control? | 426 |
| Competitive Restriction Exception (CUAD) | Does the clause mention exceptions or carveouts to Non-Compete, Exclusivity and No-Solicit of Customers? | 226 |
| Covenant Not To Sue (CUAD) | Does the clause mention if a party is restricted from contesting the validity of the counterparty's ownership of intellectual property? | 318 |
| Effective Date (CUAD) | Does the clause mention when the contract becomes effective? | 246 |
| Exclusivity (CUAD) | Does the clause mention an exclusive dealing commitment with the counterparty? | 770 |
| Expiration Date (CUAD) | Does the clause mentions a date when the contract's term expires? | 892 |
| Governing Law (CUAD) | Does the clause mentions which state/country's laws govern interpretation of the contract? | 910 |
| Insurance (CUAD) | Does the clause mention a requirement for insurance? | 1040 |
| Ip Ownership Assignment (CUAD) | Does the clause mention if intellectual property created by one party become the property of the counterparty? | 590 |
| Irrevocable Or Perpetual License (CUAD) | Does the clause describe a license grant that is irrevocable or perpetual? | 300 |
| Joint Ip Ownership (CUAD) | Does the clause provide for joint or shared ownership of intellectual property between the parties to the contract? | 198 |
| License Grant (CUAD) | Does the clause describe a license granted by one party to its counterparty? | 1430 |
| Liquidated Damages (CUAD) | Does the clause award either party liquidated damages for breach or a fee upon the termination of a contract (termination fee)? | 226 |
| Minimum Commitment (CUAD) | Does the clause specifies a minimum order size or minimum amount or units pertime period that one party must buy from the counterparty? | 778 |

| Task | Description/Intent | Size |
|---|---|---|
| No-Solicit Of Employees (CUAD) | Does the clause restricts a party's soliciting or hiring employees and/or contractors from the counterparty, whether during the contract or after the contract ends (or both). | 150 |
| Non-Compete (CUAD) | Does the clause restrict the ability of a party to compete with the counterparty or operate in a certain geography or business or technology sector? | 450 |
| Non-Disparagement (CUAD) | Does the clause require a party not to disparage the counterparty? | 108 |
| Non-Transferable License (CUAD) | Does the clause limit the ability of a party to transfer the license being granted to a third party? | 558 |
| Notice Period To Terminate Renewal (CUAD) | Does the clause requires a notice period to terminate renewal? | 234 |
| Post-Termination Services (CUAD) | Does the clause subject a party to obligations after the termination or expiration of a contract, including any post-termination transition, payment, transfer of IP, wind-down, last-buy, or similar commitments? | 816 |
| Renewal Term (CUAD) | Does the clause mention a renewal term for after the initial term expires? | 398 |
| Revenue-Profit Sharing (CUAD) | Does the clause require a party to share revenue or profit with the counterparty for any technology, goods, or services? | 784 |
| Rofr-Rofo-Rofn (CUAD) | Does the clause provide a party with a right of first refusal? | 698 |
| Source Code Escrow (CUAD) | Does the clause requires one party to deposit its source code into escrow with a third party, which can be released to the counterparty upon the occurrence of certain events (bankruptcy, insolvency, etc.)? | 126 |
| Termination For Convenience (CUAD) | Does the clause state that one party can terminate this contract without cause (solely by giving a notice and allowing a waiting period to expire)? | 442 |
| Uncapped Liability (CUAD) | Does the clause state that a party's liability is uncapped upon the breach of its obligation in the contract? | 302 |
| Volume Restriction (CUAD) | Does the clause describe a fee increase or consent requirement if one party's use of the product/services exceeds certain threshold? | 328 |
| Warranty Duration (CUAD) | Does the clause mentions the duration of any warranty against defects or errors in technology, products, or services provided under the contract? | 326 |
| BU (Learned Hands) | Does the text discuss issues relating to business or intellectual property? | 200 |
| CO (Learned Hands) | Does the text discuss issues relating to courts and lawyers? | 194 |
| CR (Learned Hands) | Does the text discuss issues relating to criminal issues? | 644 |
| ES (Learned Hands) | Does the text discuss issues relating to estates or wills? | 182 |
| FA (Learned Hands) | Does the text discuss issues relating to family or divorce? | 794 |

| Task | Description/Intent | Size |
|---|---|---|
| HE (Learned Hands) | Does the text discuss issues relating to health? | 248 |
| HO (Learned Hands) | Does the text discuss issues relating to housing? | 1270 |
| MO (Learned Hands) | Does the text discuss issues relating to payments or debt? | 740 |
| TO (Learned Hands) | Does the text discuss issues relating to accidents or harassment? | 454 |
| TR (Learned Hands) | Does the text discuss issues relating to cars or traffic? | 516 |
| WO (Learned Hands) | Does the text discuss issues relating to employment or job? | 726 |

Table 6: Science tasks

| Task | Description/Intent | Size |
|---|---|---|
| Agonist (Chemprot) | Does the sentence describe an agonist relationship? | 896 |
| Antagonist (Chemprot) | Does the sentence describe an antagonist relationship? | 1330 |
| Downregulator (Chemprot) | Does the sentence describe a downregulator relationship? | 3038 |
| Part_of (Chemprot) | Does the sentence describe a part-of relationship? | 1210 |
| Regulator (Chemprot) | Does the sentence describe a regulator relationship? | 2876 |
| Substrate (Chemprot) | Does the sentence describe a substrate relationship? | 2384 |
| Upregulator (Chemprot) | Does the sentence describe an upregulator relationship? | 2404 |
| Background (RCT) | Does the sentence describe background on the study? | 2000 |
| Conclusions (RCT) | Does the sentence state a conclusion? | 2000 |
| Methods (RCT) | Does the sentence describe a scientific experimental method? | 2000 |
| Objective (RCT) | Does the sentence describe the goal of the study? | 2000 |
| Results (RCT) | Does the sentence describe experimental results? | 2000 |

Table 7: General domain tasks

| Task | Description/Intent | Size |
|---|---|---|
| Business (AGNews) | Does the article discuss business news? | 2000 |
| Sports (AGNews) | Does the article discuss sports news? | 2000 |
| Technology (AGNews) | Does the article discuss technology news? | 2000 |
| World (AGNews) | Does the article discuss global affairs or world events? | 2000 |
| Civil Comments | Does the sentence contain toxic or hateful content? | 2500 |
| Album (DBPedia) | Is the entity discussed in the sentence an example of a album? | 2000 |
| Animal (DBPedia) | Is the entity discussed in the sentence an example of a animal? | 2000 |
| Artist (DBPedia) | Is the entity discussed in the sentence an example of a artist? | 2000 |
| Athlete (DBPedia) | Is the entity discussed in the sentence an example of a athlete? | 2000 |
| Building (DBPedia) | Is the entity discussed in the sentence an example of a building? | 2000 |
| Company (DBPedia) | Is the entity discussed in the sentence an example of a company? | 2000 |
| Educational institution (DBPedia) | Does the sentence discuss a school, university, or college? | 2000 |

| Task | Description/Intent | Size |
| --- | --- | --- |
| Film (DBPedia) | Is the entity discussed in the sentence an example of a film? | 2000 |
| Mean of transportation (DBPedia) | Does the sentence discuss a car, ship, train, or plane? | 2000 |
| Natural place (DBPedia) | Is the entity discussed in the sentence an example of a natural landscape or environment? | 2000 |
| Office holder (DBPedia) | Is the entity discussed in the sentence an example of a office holder? | 2000 |
| Plant (DBPedia) | Is the entity discussed in the sentence an example of a plant? | 2000 |
| Village (DBPedia) | Is the entity discussed in the sentence an example of a village? | 2000 |
| Written work (DBPedia) | Is the entity discussed in the sentence a writing? | 2000 |
| Architect (FewRel) | Does the text mention an architect? | 600 |
| Composer (FewRel) | Does the text mention a musical composer? | 600 |
| Country (FewRel) | Does the text mention a country? | 600 |
| Developer (FewRel) | Does the text mention development? | 600 |
| Director (FewRel) | Does the text mention a film director? | 600 |
| Distributor (FewRel) | Does the text mention a film distributor? | 600 |
| Father (FewRel) | Does the text mention a father? | 600 |
| Genre (FewRel) | Does the text mention the genre of a song or artist? | 600 |
| Instrument (FewRel) | Does the text mention an instrument? | 600 |
| League (FewRel) | Does the text mention a sports competition, league, or division? | 600 |
| Military Branch (FewRel) | Does the text mention a military branch? | 600 |
| Movement (FewRel) | Does the text mention an art movement? | 600 |
| Occupation (FewRel) | Does the text mention a professional occupation? | 600 |
| Participating Team (FewRel) | Does the text mention a sports team? | 600 |
| Platform (FewRel) | Does the text mention an online platform? | 600 |
| Sibling (FewRel) | Does the text mention a sibling? | 600 |
| Successful Candidate (FewRel) | Does the text mention an election winner? | 600 |
| Taxonomy Rank (FewRel) | Does the text mention a taxonomy class of animals? | 600 |
| Tributary (FewRel) | Does the text mention a tributary? | 600 |
| Winner (FewRel) | Does the text mention a competition winner? | 600 |
| YouTube Spam Detection | Does the comment ask the user to check out another video? | 1836 |

# E  Prompts

We construct a prompt for each task by randomly selecting three examples of each class to use as in-context demonstrations [51]. We manually defined "instructions" for each task. An example of an abridged prompt is shown in Figure 3.

```
┌─────────────────────────────┐
│   Instruction Classification │
│            Prompt            │
└─────────────────────────────┘

Is the entity discussed in the sentence an example of an animal?

Sentence: " Hugh Christopher Chris Brown is a Canadian singer-
songwriter and multi-instrumentalist."
Label: No

Sentence: " Forficula auricularia the common earwig or European earwig
is an omnivorous insect in the family Forficulidae. The European earwig
survives in a variety of environments and is a common household insect
in North America. The name earwig comes from a false superstition that
these insects crawl into human ears and enter the brain; in fact they
are harmless to humans."
Label: Yes

Sentence: " Northern Dancer (May 27 1961 – November 16 1990) was a
Canadian-bred Thoroughbred racehorse who won the Kentucky Derby and
Preakness Stakes and who became the most successful sire of the 20th
Century. The National Thoroughbred Racing Association calls him one of
the most influential sires in Thoroughbred history.A bay colt Northern
Dancer was by Nearctic and his dam Natalma was by Native Dancer. In
1952 Edward P."
Label:
```

Figure 3: An example of the instruction classification prompt for the dbpedia_animal task, with two in-context demonstrations. Here, the task instructions are in red, the in-context demonstrations are in blue, and the sample for which we want a label is in green.

## F  Synthetics

We conduct synthetic experiments which provide additional insights on EMBROID.

For our setup, we create two equal clusters of data of 500 points each, $C_1$ and $C_2$, in $\mathbb{R}^2$. We assign labels to the points in each cluster i.i.d. according to a probability $p$, where $\Pr(y = 1 | x \in C_1) = p$ and $\Pr(y = 1 | x \in C_2) = 1 - p$. When $p = 0.5$, both the clusters have a uniform label distribution (non-smooth) while $p = 1$ ensures each cluster has one class (smooth). We set $k = 20$, $\tau_+ = P(\lambda_i = 1)$ and $\tau_- = P(\lambda_i = -1)$.

**Improvement over weak supervision**. We show that EMBROID offers improvement over methods that only use $\boldsymbol{\lambda}$. We fix $p = 0.8$ and $\beta_i = \Pr(\lambda_i = y) = 0.6$ for each $i \in [m]$. We compare EMBROID against the standard weak supervision approach from [14], which requires $m \geq 3$, in Figure 4a.

**Smoothness**. EMBROID's performance depends on the smoothness of the embedding as defined in eq. (4). We consider one LM prediction $m = 1$ and vary the smoothness $p$ from 0.5 to 1.0 and generate predictions using $\beta_i = 0.6$. Figure 4b exhibits that EMBROID's accuracy is positively correlated with the embedding smoothness.

**Base prediction accuracy**. Finally, we show that EMBROID's performance depends on the base prediction accuracy, $\beta_i$. We consider $m = 1$ and set $p = 0$, $\beta_i = 0.8$. We use a parameter $\rho$ to denote the probability that $\lambda_i$ is incorrect on points in cluster $C_1$. As $\rho$ varies from 0 to 1, the predictions of $\lambda_i$ become biased towards 1 and effectively reduces $\beta_i$ to 0.5. In Figure 4c, we observe that as the base prediction accuracy decreases, EMBROID's performance decreases and eventually goes below the base LM performance.

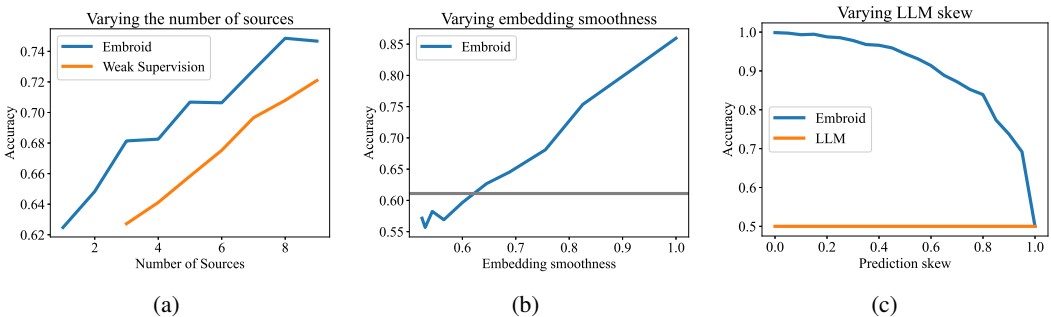

Figure 4: Synthetic experiments. (a) Comparison of EMBROID to weak supervision by varying number of LLM sources. Increasing sources consistently improves both EMBROID and WS and the gains remain constant. (b) EMBROID's performance as embedding smoothness (eq. (4)) varies. EMBROID's accuracy linearly improves as a function of embedding smoothness. (c) EMBROID's performance with varying probability of LLM being incorrect in $C_1$. As the LM becomes incorrect, the cluster becomes less homogeneous and this degrades EMBROID's performance.

## G  Ablation

We perform additional ablations of EMBROID. We focus on two aspects:

- The role of $\tau^-/\tau^+$.
- The impact of weak supervision in combining $\lambda_{\mathsf{sm},j}$.

### G.1  Ablation over $\tau$

In our experiments, we set $\tau_i^+ = \tau_i^- = \mathbb{E}[\lambda_i]$, or the average vote of source $\lambda_i$. This has the following effect on the neighborhood vote $\lambda_{\mathsf{sm},j}[i]$ for source $\lambda_i$ under $E_j$:

- When the average vote for a source $\lambda_i$ in a neighborhood under $E_j$ for $x$ is more negative than the average overall vote for a source, then $\lambda_{\mathsf{sm},j}[i](x) = -1$.

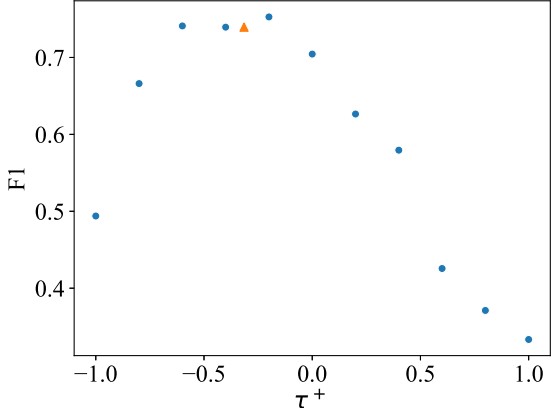

Figure 5: We analyze how F1 changes for different settings of $\tau_i^+/\tau_i^-$ for GPT-JT on a task. Observe that setting $\tau_i^+ = \tau_i^- = \mathbb{E}[\lambda_i]$ (denoted as the orange triangle) produces close-to-optimal performance.

| LM | Base prompt | Majority vote aggregation | EMBROID |
|---|---|---|---|
| j1-jumbo | 0.498 | 0.569 | 0.604 |
| openai_text-davinci-003 | 0.806 | 0.844 | 0.855 |
| bloom-7b1 | 54.7 | 61.2 | 64.7 |
| opt-6.7b | 48.2 | 56.1 | 59.8 |
| GPT-JT-6B-v1 | 67.8 | 73.2 | 75.1 |

Table 8: We evaluate how EMBROID compares to a majority vote aggregation over neighborhood vote vectors. We report macro-F1 average across all tasks and prompts, mirroring results in Table 1.

- When the average vote for a source $\lambda_i$ in a neighborhood under $E_j$ for $x$ is more positive than the average overall vote for a source, then $\lambda_{\mathsf{sm},j}[i](x) = 1$.

In general, we find that this setting provides good performance, while requiring no additional tuning or validation. For example, the Figure 5 below compares a setting of $\tau_i^+/\tau_i^-$ against the F1 score for GPT-JT on ag_news_business.

## G.2 Role of weak supervision aggregation

We quantify the extent to which performance gains are derived from (1) the computation of $\boldsymbol{\lambda}_{\mathsf{sm}}$, as opposed to (2) the use weak-supervision [14] to combine $\boldsymbol{\lambda}_{\mathsf{sm}}$ and $\boldsymbol{\lambda}$. Specifically, we replace Equation 3 with a simple majority vote classifier which combines the original prediction with the computed neighborhood votes.

## G.3 Choice of embeddings

We provide additional discussion on how practitioners may go about choosing embeddings for EMBROID. At the outset, we note that selecting/evaluating embeddings is particularly difficult in our regime, where practitioners do not have access to labeled data. Importantly, this motivates Embroid's "mixture of embeddings" approach. Previous methods (e.g., Liger [7]) operate on only a single embedding space. Thus, practitioners must be certain that the embedding space chosen is good for a task, and a poorly chosen embedding space will reduce performance. Because EMBROID ensembles different embedding spaces, practitioners can be less certain in the quality of any individual embedding space.

A natural strategy may be to start with an initial large set of candidate embeddings, and drop embeddings according to task-specific measurements (akin to feature selection). Several criteria may be used to determine which embeddings to maintain:

- Looking at the MLM loss of the embedding model (e.g. BERT) on the task data. Prior literature on domain specificity has suggested that this may be a promising heuristic [71].

- Looking to the performance of embedding models on other related tasks may also be helpful [38, 71].

- Looking at the extent to which a potential embedding space generates embeddings which are geometrically similar to embeddings already included in Embroid. If the embedding space is similar, then the additional value of incorporation is low.

The number of embeddings used presents a bias-variance tradeoff. Under Proposition 5.1, increasing the number of embedding spaces (1) increases the variance due to estimation error, but lowers (2) the conditional entropy of $H(y|\lambda)$. Precisely characterizing this tradeoff is challenging because the variance is an upper bound, and $H(y|\lambda)$ can only be estimated. However, these bounds/estimates can be used to derive heuristic stopping rules: for instance, add embedding spaces until the marginal decrease in the conditional entropy is less than the upper bound on the marginal increase in variance.

# H Experiments

## H.1 Implementation details

**Compute** Inference for API-access models (e.g., GPT-3.5 and J1-Jumbo) were run using the HELM API [38]. Inference for open source models (OPT, GPT-JT, and Bloom) were run using the Manifest library [49] on 40GB A100 NVIDIA GPU machines.

**Hyperparameters** EMBROID was run with $k = 10$, $\tau_i^+ = P(\lambda_i = 1)$, and $\tau_i^- = P(\lambda_i = -1)$.

## H.2 Space demand

We provide additional information on the space requirements for EMBROID. First, we clarify that EMBROID requires only one large model. This model can be on the order of GPT-3 (176B parameters), or a much smaller open-source equivalent like GPT-JT (7B parameters). Next, EMBROID requires embeddings from 2-3 additional models. However, as our results show, these models can be relatively small. For instance, we rely on BERT-base and SentenceBERT models–both of which are approximately 110 million parameters. Finally, the latent variable graphical model has 2 parameters for class prior and 1+N parameters per LLM prediction, where N is the number of embedding spaces. So, when EMBROID is run on a single LLM's predictions with two embedding spaces, the total number of parameters is 2 + (1+2)*1 = 5.

## H.3 Runtime

In terms of time complexity, we observe that computing LLM predictions is the largest bottleneck. On a dataset of 1800 samples for instance:

- Computing predictions on GPT-3.5 takes 1440 seconds (24 minutes).
- Computing embeddings using BERT takes 5 seconds.
- Computing nearest-neighbors using the (unoptimized) scikit-learn library takes 3 seconds.
- Solving the Embroid graphical model takes less than a second.

**API-model tasks** Due to cost constraints, we study the API access models (GPT-3.5 and J1-Jumbo) on a subset of tasks. These are:

- `ag_news_world`
- `ag_news_sports`
- `dbpedia_educational institution`
- `dbpedia_athletechemprot_regulator`
- `chemprot_upregulator`
- `rct_objective`
- `rct_methods`
- `CUAD_Audit Rights`
- `CUAD_Non-Compete`
- `learned_hands_HE`
- `learned_hands_HO`
- `few_rel_architect`
- `few_rel_league`

## H.4 Robustness across models

We provide the results for each LM on each task as CSV files in the supplemental attachment. We visualize the improvements in Table 1 below, by plotting the original prompt performance on the x-axis, and the performance after EMBROID on the y-axis (Figure 6).

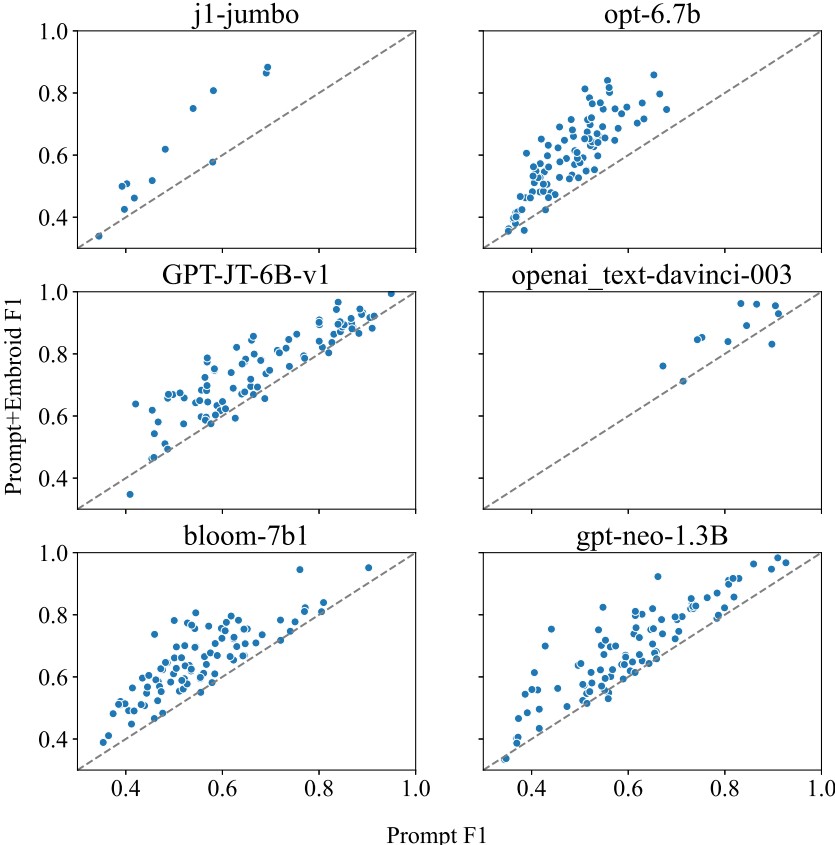

Figure 6: We visualize the improvement from EMBROID over the base prompt for each model's tasks. All models except for gpt-neo-1.3B were run thrice per task, with each run using different in-context samples for the prompt. Each dot corresponds to a task. The x-axis measures the average macro F1 of the base prompt, and the y-axis measures the average macro F1 of EMBROID (across all runs). Because GPT-3.5 and J1-Jumbo are studied on a subset of 12 tasks, there are fewer dots in the plot.

## H.5 Comparison to AMA

We visualize the improvements in Table 2 below, by plotting the performance of AMA on the x-axis, and the performance of EMBROID-3 on the y-axis (Figure 7).

## H.6 Chain-of-thought experiments

We compare EMBROID to chain-of-thought (CoT) [64] on the following tasks:

- `ag_news_business`
- `ag_news_sports`
- `CUAD_Affiliate License-Licensee`
- `CUAD_Audit Rights`
- `dbpedia_album`
- `dbpedia_building`
- `few_rel_architect`
- `few_rel_country`
- `learned_hands_HO`
- `learned_hands_MO`

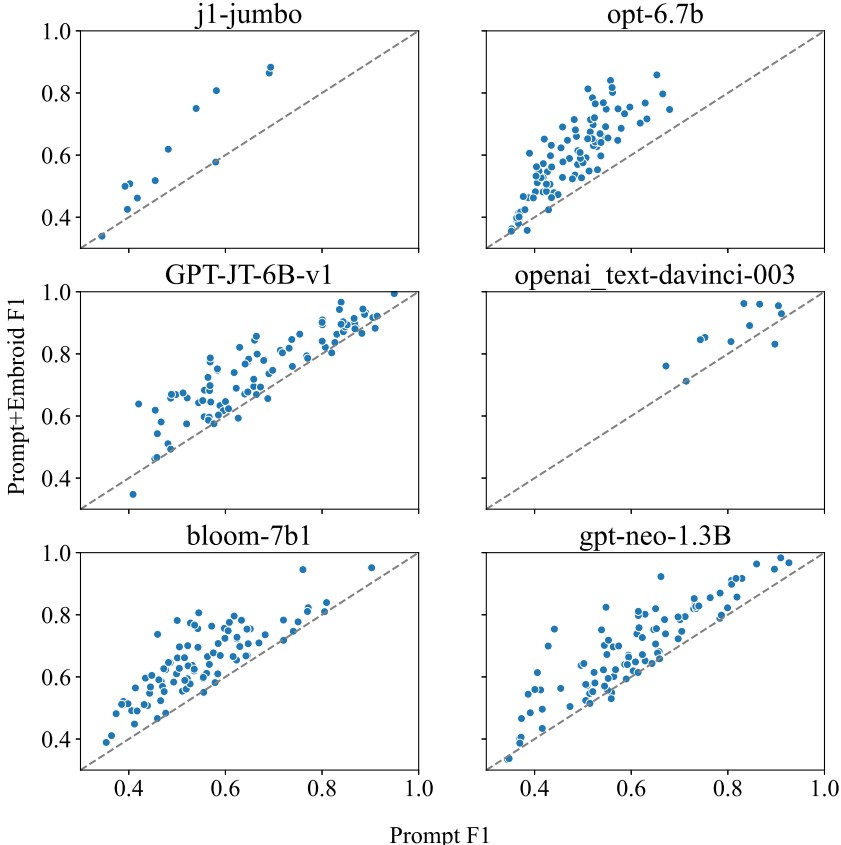

Figure 7: We visualize the improvement from EMBROID over AMA [2] for each model's tasks. Each dot corresponds to a task. The x-axis measures the average macro F1 of AMA, and the y-axis measures the average macro F1 of EMBROID. Because GPT-3.5 and J1-Jumbo are studied on a subset of 12 tasks, there are fewer dots in the plot.

We manually write logical chains for each prompt. Because CoT primarily succeeds on "large" models [64], we focus our experiments on GPT-3.5.

## H.7  Comparison to GPT-4

GPT-4 [48] was made publicly available after this work was completed. Nonetheless, we offer preliminary results evaluating EMBROID on predictions produced by GPT-4. We evaluated EMBROIDEmbroid+GPT-4 on a subset of 14 tasks. These are:

- `ag_news_world`
- `ag_news_sports`
- `dbpedia_educational institution`
- `dbpedia_athletechemprot_regulator`
- `chemprot_upregulator`
- `rct_objective`
- `rct_methods`
- `CUAD_Audit Rights`
- `CUAD_Non-Compete`
- `learned_hands_HE`
- `learned_hands_HO`

- `few_rel_architect`
- `few_rel_league`

We note that overall, GPT-4 appears to saturate our evaluated tasks, scoring greater than 95% on 5 of them. Given that we evaluate on publicly accessible benchmarks, this could be explained by leakage into GPT-4's pre-training data. We find that EMBROID is strictly better than the base prompt on 42% of tasks, and effectively equivalent or better (i.e., no worse than 1 point F1) on 73% of tasks. On two tasks Embroid improves by 4+ points.

Though EMBROID 's absolute gains over base prompts may diminish for stronger base LMs like GPT-4, our evaluation and findings focused on open-source models because of their practical feasibility. Such models are cheaper and better suited for sensitive domains, as they allow data to remain on-premises. This is a particularly salient concern for applications in medicine, law, or government. On these models, we found that EMBROID offered substantial gains, with a win-rate > 88% and an average improvement > 7 points F1.

## H.8   Comparison to self-consistency

We evaluate EMBROID against self-consistency [63] over GPT-3.5, and find that (1) it can provide competitive performance to self-consistency, and (2) that it can improve predictions generated from self-consistency.

We study a subset of five tasks: `ag_news_business`, `CUAD_Audit Rights`, `dbpedia_album`, `few_rel_architect`, `learned_hands_HO`. For self-consistency, we generate 5 predictions per sample, using a temperature of 0.7. For each task, we consider a "base" prompt which does not use chain-of-thought, but otherwise uses the same task instructions and in-context demonstrations. For 4/5 tasks, EMBROID applied to the output of the base prompt outperforms self-consistency. On all tasks, EMBROID applied to the predictions generated from self-consistency improve performance, by a median of six points.

We additionally note one important distinction between self-consistency and EMBROID. While self-consistency requires multiple predictions from a LM for a single sample (thus accruing additional cost in hardware usage or API calls), Embroid requires only one prediction per-sample.

## H.9   Multi-class experiments

We present initial results evaluating EMBROID in the multi-class setting. We create a multi-class version of the contract classification task by combining five different binary classification datasets: `CUAD_Audit right`, `CUAD_Governing law`, `CUAD_Non-compete`, `CUAD_Volume restriction`, and `CUAD_Expiration date`. The total size of this dataset is 1858, with the largest class (602 samples) occurring nearly 3.5x as much as the smallest class (165 samples).

We evaluate a one-vs-all version of EMBROID, where EMBROID is applied five times—each time predicting one of the classes. A final class prediction is generated by comparing each of the class-specific EMBROID classifier's predicted probability for the class, and choosing the class with the highest probability. Overall, we find that this variant of EMBROID improves the quality of the base prompt (by 4 points on balanced-accuracy and 1.6 points on macro-F1).

